# Experimental Evaluation of Sensor Fusion of Low-Cost UWB and IMU for Localization under Indoor Dynamic Testing Conditions

**DOI:** 10.3390/s22218156

**Published:** 2022-10-25

**Authors:** Chengkun Liu, Tchamie Kadja, Vamsy P. Chodavarapu

**Affiliations:** Department of Electrical and Computer Engineering, University of Dayton, 300 College Park, Dayton, OH 45469, USA

**Keywords:** ultra-wideband (UWB), inertial measurement unit (IMU), sensor fusion, Kalman filter, system localization

## Abstract

Autonomous systems usually require accurate localization methods for them to navigate safely in indoor environments. Most localization methods are expensive and difficult to set up. In this work, we built a low-cost and portable indoor location tracking system by using Raspberry Pi 4 computer, ultra-wideband (UWB) sensors, and inertial measurement unit(s) (IMU). We also developed the data logging software and the Kalman filter (KF) sensor fusion algorithm to process the data from a low-power UWB transceiver (Decawave, model DWM1001) module and IMU device (Bosch, model BNO055). Autonomous systems move with different velocities and accelerations, which requires its localization performance to be evaluated under diverse motion conditions. We built a dynamic testing platform to generate not only the ground truth trajectory but also the ground truth acceleration and velocity. In this way, our tracking system’s localization performance can be evaluated under dynamic testing conditions. The novel contributions in this work are a low-cost, low-power, tracking system hardware–software design, and an experimental setup to observe the tracking system’s localization performance under different dynamic testing conditions. The testing platform has a 1 m translation length and 80 μm of bidirectional repeatability. The tracking system’s localization performance was evaluated under dynamic conditions with eight different combinations of acceleration and velocity. The ground truth accelerations varied from 0.6 to 1.6 m/s^2^ and the ground truth velocities varied from 0.6 to 0.8 m/s. Our experimental results show that the location error can reach up to 50 cm under dynamic testing conditions when only relying on the UWB sensor, with the KF sensor fusion of UWB and IMU, the location error decreases to 13.7 cm.

## 1. Introduction

The development of concepts for the accurate navigation of robots and autonomous systems in global positioning system (GPS)-denied indoor environments is an area of active research [1,2,3]. Previously, research groups have relied on a variety of signals and systems to create accurate navigation solutions for autonomous systems to operate in GPS-denied indoor environments including, visual cameras [4], radio frequency (RF) beacons [5], inertial sensors [6], ultrasound [7], RADAR [8], LiDAR [9], and thermal cameras [10]. Each of the above methods has its own strengths and weakness. Thus, sensor fusion is applied to combine several of the above methods to meet the requirements of long-term and accurate navigation [11].

In indoor environments, radio frequency (RF) beacons are usually used to provide location services. The location accuracy of the RF method depends on the accuracy of the distance measurement between a tag and an anchor. A tag is a RF device that is installed on a moving object. An anchor is a RF device that has a predetermined fixed location. Wi-Fi and Bluetooth use received signal strength indicator(s) (RSSI) used to measure distance [12]. The signal strength varies according to the inverse square of the distance. There is a math equation to calculate the distance from the RSSI. However, RSSI is influenced by obstacles and this will reduce the accuracy of the distance measurement.

UWB [13] is a specially designed RF signal that enables precise, secure, real-time measurements of location, distance, and direction. The bandwidth of UWB is greater than 500 MHz. Due to the inverse relationship between time and bandwidth, the time-durations of the UWB signals are extremely short (around 2 ns) and the rising edge is very steep. This brings at least two benefits, the first is that the time resolution of the UWB signal is high. UWB relies on time-of-flight (ToF) to calculate the distance. A more accurate time measurement results in a more accurate distance measurement. Secondly, UWB signals are robust and resistant to multipath interference. The short time duration can allow the arrival of the reflected pulse to be separated and filtered at the receiver.

UWB is standardized by IEEE 802.15.4a/z standards and enables high accuracy ranging and positioning. The range of UWB is between 100 and 200 m. This allows for a small amount of UWB anchors to cover a large area. However, relying exclusively on UWB does not generate a satisfactory localization performance when the system is in motion. Hence, it is used in combination with other sensors, such as an IMU, to reduce latency and improve the accuracy of positioning. IMUs are typically available with six degrees of freedom, which combine three-orthogonal axis accelerometers and three-orthogonal axis gyroscopes. Many research groups have implemented the sensor fusion of UWB and IMU for localization, as listed in Table 1. They rely on KF-based methods and assume the noise follows Gaussian distribution. KF is used for linear systems, while the extended Kalman filter (EKF) and unscented Kalman filter (UKF) are used for non-linear systems.

The general sensor fusion algorithm of UWB and IMU relies on the Bayesian filtering framework, which is based on Markov assumption. Only one previous state was considered to calculate the current state. An optimization-based sensor fusion framework [24] is proposed to make better usage of the logged data. More previous states can be used for the calculation of the current state by a sliding trajectory window. A cost-effective wearable localization system [25] aims to provide affordable indoor location services by relying on low-cost–low-power sensors. Another research direction focuses on finding the optimum installation of UWB anchors. Different anchor installation positions will influence the localization accuracy, this paper analyzes the relationship between the anchor’s installation and the localization accuracy [26]. Manually measuring the anchor’s position is time-consuming; an anchor’s self-localization algorithm was proposed to ease the installation procedure and reduce the time to deploy the localization system [27]. Other research teams have worked on using a single anchor [28] or the minimum amount of anchors [29] for localization. Particle filter [30,31,32] is another popular sensor fusion algorithm used to fuse UWB/IMU data for location purposes. With artificial intelligence and machine learning becoming more popular, some researchers are working on neural networks and machine learning to achieve high localization accuracy and activity recognition [33,34,35].

Autonomous systems navigating in indoor environments are subject to diverse motion conditions [36,37,38], which requires that the localization performances also be evaluated under such dynamic scenarios. From Table 1, prior works have reported accuracies as high as 0.03 m. VICON [39] is a high-precision optical tracking system and can provide a centimeter-level ground truth trajectory, but its cost is high. Placing a tracking device on a mobile robot is less expensive, but the generated ground truth is also less accurate. The evaluation of localization performance under dynamic testing conditions requires relevant ground truth data for accurate assessment. The ground truth data include the ground truth trajectories, accelerations, and velocities for the system. Prior works have mostly considered the ground truth trajectory [19], but have not considered ground truth acceleration and velocity. We believe that this is a drawback, as even if a system has the same ground trajectory, having different velocities and accelerations changes the motion conditions.

In this work, we focused on evaluating the localization performance of our tracking system under dynamic testing conditions. To this end, we developed a low-cost–low-power tracking system using off-shelf components: Raspberry Pi 4 (computer), DWM1001 (UWB), and BNO055 (IMU). The accuracy and performance of the tracking system could be improved by using more expensive and higher-performance sensors. Both the data logging software and the KF sensor fusion algorithm were developed to record UWB and IMU data and estimate the tracking system’s location. The KF measurement update procedure requires linearly mapping the state vector into the measurement space. Decawave provides the positioning and networking stack (PANS) library and this library contains the location engine (LE) function. UWB LE can directly output the 3D coordinates, which meet the linear mapping requirements during the KF measurement update procedure.

Our work provides five contributions. (1) We designed and built a low-cost, low-power, and portable UWB/IMU data acquisition (DAQ) system hardware and software based on Raspberry Pi and Linux. It has a Wi-Fi connection and a file system to store logged data. (2) We used DWM1001 instead of its predecessor DWM1000. DWM1001 has an onboard system-on-chip (SoC) Nordic nRF52832, which is based on a 64 MHz ARM Cortex M4. The PANS firmware supports LE to directly output the tag’s 3D location coordinates. This enables us to linearly map the state vector into the measurement space. In this way, we can use the KF method for sensor fusion of UWB and IMU. Most of the previous works used ranging instead of LE and they had to use the EKF or UKF methods, which required either calculating the Jacobian matrix or dealing with sigma points. Our KF algorithm reduces time complexity compared with the previous EKF and UKF methods. (3) We used a highly accurate linear stage for the UWB/IMU tracking device to move at a desired target speed and acceleration, while most previous papers placed the tracking device on a person or a robot, which is not accurate compared with the linear stage. Our testing platform generates not only the ground truth trajectories but also the ground truth accelerations and velocities. These ground truth values are used to evaluate our tracking system’s dynamic performance. (4) Our tracking system’s localization performance is tested under diverse motion conditions with eight different combinations of acceleration and velocity. The experiment result shows that UWB’s localization performance degrades when the system is in motion, and thus, an IMU is needed to compensate for the UWB’s performance degradation under dynamic motion conditions. (5) Our KF design and the custom-built DAQ software do not assume constant sampling frequencies. The data from UWB/IMU need to be logged with timestamps in our experiments. The KF algorithm also needs to recalculate the Δt between two samples for its prediction and update procedure. In comparison, most previous papers assume a constant sampling frequency, which may not hold in real-world applications.

This paper is organized as follows. Section 2 introduces some background knowledge about UWB localization and IMU strapdown navigation. Section 3 explains the system design of our tracking system, test platform setup, and the KF sensor fusion algorithm. Section 4 shows the experiment results of localization performance under dynamic testing conditions. Section 5 summarizes our work and proposes some ideas for future research.

## 2. Background Knowledge

### 2.1. Impulse-Radio UWB Localization

UWB allows precise, secure, real-time measurement of location, distance, and direction [40]. It also supports two-way communications. UWB signals use time-of-flight (ToF) to calculate distance. In a UWB real-time locating system (RTLS), UWB anchors have known fixed locations and UWB tag(s) are placed on an object for tracking. The RTLS requires three or more anchors to work accurately. UWB ranging can be implemented in two different ways: two-way ranging (TWR) and time difference of arrival (TDoA). After the ranging, the location of the tag can be determined by trilateration.

Assume that the UWB tag to be tracked is at an unknown location (x,y,z), and the *i*th anchor is at a known location $\left(x_{i},y_{i},z_{i}\right)$. The measured distance between them is $d_{i}$ and is given from Equation (Equation 1),
(1)$$d_{i}=\sqrt{{\left(x_{i}-x\right)}^{2}+{\left(y_{i}-y\right)}^{2}+{\left(z_{i}-z\right)}^{2}},$$

The location of the tag can be calculated from three measured distances by solving the system of Equations (Equation 2) [41],
(2)$$\left\{\begin{array}{c} d_{1}^{2}={\left(x_{1}-x\right)}^{2}+{\left(y_{1}-y\right)}^{2}+{\left(z_{1}-z\right)}^{2}\\ d_{2}^{2}={\left(x_{2}-x\right)}^{2}+{\left(y_{2}-y\right)}^{2}+{\left(z_{2}-z\right)}^{2}\\ d_{3}^{2}={\left(x_{3}-x\right)}^{2}+{\left(y_{3}-y\right)}^{2}+{\left(z_{3}-z\right)}^{2}, \end{array}\right.$$

### 2.2. IMU and Strapdown Inertial Navigation

IMUs are generally not suitable for long-term navigation and require aiding from other signals or systems, such as GPS, UWB, or visible cameras to bind their errors [42]. In this work, we will refer to the IMU’s frame as the body frame and the fixed reference frame as the global frame. The body frame and global frame do not always point in the same direction in the real world. Thus, the IMU measurements cannot be used directly if the body frame is not aligned with the global frame. There are two different methods to solve this problem, the first one is to mount the IMU on a stable platform that is isolated from the external rotation, such as a three-axis gimbal. This installation enables the body frame and the global frame to always point in the same direction. The second solution is to rigidly install IMU onto the object. This setting is called strapdown inertial navigation. When the body frame and global frame are pointing in different directions, the measurements need to be projected into the global frame, and then remove the gravitational acceleration [43]. One can use a direction cosine matrix, a 3×3 rotation matrix C(t) to represent orientation, a vector $a_{g}\left(t\right)={\left(a_{gx}\left(t\right),a_{gy}\left(t\right),a_{gz}\left(t\right)\right)}^{T}$ to represent acceleration in global frame, and a vector $a_{b}\left(t\right)={\left(a_{bx}\left(t\right),a_{by}\left(t\right),a_{bz}\left(t\right)\right)}^{T}$ to represent the acceleration in body frame. The acceleration in the body frame can be projected into the global frame using Equation (Equation 3) [44],
(3)$$a_{g}\left(t\right)=C\left(t\right)a_{b}\left(t\right),$$

An accelerometer can be modeled as a mass suspended by three orthogonal directions of spring. Mass is subject to the Earth’s gravity. Thus, an accelerometer always measures gravitational acceleration. When the body frame is aligned with the global frame, the measurement of acceleration at rest is $a_{b}={\left(0,0,g\right)}^{T}$, and free fall is $a_{b}={\left(0,0,0\right)}^{T}$. It is easy to remove the gravitational acceleration when two frames are aligned with each other. When the body frame and global frame are pointing in different directions, the measurements need to be projected into the global frame, then remove the gravitational acceleration. Gravitational acceleration and different coordinate frames are shown in Figure 1.

## 3. Methods and Test Platform

We built the hardware and DAQ software for a UWB/IMU tracking system, set up the test platform, and designed the KF sensor fusion algorithm to estimate the state vector. The tracking system’s localization performance was evaluated while the test platform was under diverse motion conditions. This Section describes the three subsystems and compares the proposed KF method with EKF and UKF methods. The materials required to conduct the experiment are listed in Table 2.

### 3.1. UWB and IMU Tracking System Design

The system design of the UWB and IMU tracking system is shown in Figure 2. The UWB sensors included one tag and four UWB anchors, which were attached to four tripods. The anchors and the tag shared the same hardware DWM1001, with different configurations in Decawave DRTLS Manager, which ran on an Android tablet. Raspberry Pi 4 utilized a Pi-EzConnect shield board to connect to the DWM1001-Dev board and BNO055 IMU. The tracking system had a low power consumption and was powered by batteries. The whole tracking system was portable and could be deployed easily in a short time.

The sampling frequency was not a constant number in our custom-built UWB/IMU DAQ system. Raspberry Pi ran on a Linux operating system and our DAQ software operated as a process in Linux. Because Linux is not a real-time operating system, this leads to a variable sampling frequency. The Δt between the two samples was not a constant number and it kept changing. This required logging the data from DWM1001 and BNO055 and associating a timestamp with the logged data. The Δt between two samples could be easily calculated by subtracting their timestamps. Both Decawave and Bosch provided API to the interface with their sensors. Our custom-built DAQ software used these APIs to communicate with UWB/IMU sensors and log the data on Raspberry Pi’s SD card as a comma-separated file (CSV). This file included: timestamp, x, y, z coordinates, signal quality factor, x, y, z raw accelerations, x, y, z linear accelerations, x, y, z angular speeds.

The Raspberry Pi 4 model B has 40 GPIO pins, Linux, Wi-Fi, UART, I2C, SPI, file system, and an SD card. Moreover, DWM1001’s API works well on a Linux computer and it provides both ranging and LE functions. The LE can directly output the 3D location coordinates. These features make Raspberry Pi a good choice for building our tracking the system and accessing the logged data remotely.

BNO055 is a system in a package-intelligent nine-axis absolute orientation sensor. It provides both UART and I2C interfaces and can be configured to output raw IMU data or fused data. The DWM1001 module is soldered on a DWM1001-DEV board, which provides UART, SPI, and Bluetooth interfaces. The Bluetooth interface is reserved for configuration by connecting with an Android tablet running the Decawave DRTLS manager. In the DRTLS manager, UWB’s mode type, network, location engine, and update rate can be configured. In our experimental setup, UART is selected to interface with DWM1001, and I2C is selected to interface with BNO055.

There are two different methods used to access the UWB tag location: UART generic and UART shell. In the UART shell mode, one would input a command and wait for the response. In the UART generic mode, Raspberry Pi sends out a type-length-value (TLV) request message and waits for a TLV response message. After decoding the TLV message, the location data can be logged into a CSV file automatically. We chose the UART generic mode for our tracking system. The UWB/IMU DAQ software was developed in C programming language. The logged CSV data were processed in Matlab.

### 3.2. Test Platform Setup

The Newmark motion control system was used to generate accurate linear motion and rotation. The ground truth trajectory could reach an accuracy of 80 μm. The Newmark CS-1000-1 belt drive’s linear stage specifications are shown in Table 3.

The generated ground truth contains linear trajectory, acceleration, and velocity. The CS-1000-1 linear stage was used for linear motion, the RM-3-110 motorized rotary stage was used for rotation. The controllers used for linear motion and rotation were NSC-A1 and NSC-G4. NSC-A1 was connected via USB and NSC-G4 was connected via RS232 to a personal computer (PC). This PC had Newmark’s motion control software installed. The testing platform’s target speed and acceleration time could be configured in Newmark’s motion control software. The testing platform is shown in Figure 3. The UWB–IMU tracking device was installed on this testing platform, as shown in Figure 4.

A custom-built DAQ software was developed in C programming language to log both UWB and IMU data together with timestamps. The data measured at the same moments by DWM1001 and BNO055 needed to be logged with the same timestamp. Both Decawave and Bosch provided APIs to communicate with their sensors. They were used to read/write the values of designated registers. The KF sensor fusion method was developed in Matlab to process the logged UWB and IMU data.

### 3.3. Kalman Filter Design

Our experiment was conducted in a laboratory environment and the UWB tag was in line-of-sight (LOS) with UWB anchors. We did not experience data dropouts or delays during our data logging procedure. We assumed that the system’s uncertainty followed Gaussian distribution in order to use the KF-based sensor fusion method. Although our tracking system logged all the data in 3D format, which included xyz coordinate, xyz acceleration, and xyz rotation, our testing platform only moved in 2D xy coordinate. Thus, the KF was designed in 2D instead of 3D. The 2D KF method required fewer computations, thus, it was more appropriate for deploying on an embedded computer.

The 2D KF method was applied to fuse data from UWB and IMU to estimate the state of the system. Here, ν was used to denote the noise caused by motion, and ω was used to denote the noise from measurements. Both ν and ω followed zero mean Gaussian distributions. The measurement noise ω was determined by the manufacturer of the sensor. It was considered as a constant number for simplicity. Because the Δt between two logged samples kept changing, the motion noise ν (sometimes called the process noise) also kept changing. The longer the Δt between two samples, the more uncertainty will be generated in the prediction procedure, which results in a larger variance. Both prediction and measurement update procedures can give some information about an object’s location. The state estimation is between the mean of the prediction and the mean of the measurement. The state vector of the 2D KF is given by Equation (Equation 4),
(4)$$x={\left(\begin{array}{cccccc} p_{x} & p_{y} & v_{x} & v_{y} & a_{x} & a_{y} \end{array}\right)}^{⊤},$$

After the prediction procedure of the motion model, the state vector becomes $x^{\prime }$: (5)$$x^{\prime }={\left(\begin{array}{cccccc} p_{x}^{\prime } & p_{y}^{\prime } & v_{x}^{\prime } & v_{y}^{\prime } & a_{x}^{\prime } & a_{y}^{\prime } \end{array}\right)}^{⊤},$$

Jerk is considered the noise caused by motion in our model. Jerk is the rate at which an object’s acceleration changes with respect to time. Here, $j_{x}$ represents the jerk on the *x*-axis and $j_{y}$ represents the jerk on the *y*-axis. The motion model is given by Equation (Equation 6),
(6)$$\left\{\begin{array}{c} p_{x}^{\prime }=p_{x}+v_{x}\Delta t+\frac{1}{2}a_{x}\Delta t^{2}+\frac{1}{6}j_{x}\Delta t^{3}\\ p_{y}^{\prime }=p_{y}+v_{y}\Delta t+\frac{1}{2}a_{y}\Delta t^{2}+\frac{1}{6}j_{y}\Delta t^{3}\\ v_{x}^{\prime }=v_{x}+a_{x}\Delta t+\frac{1}{2}j_{x}\Delta t^{2}\\ v_{y}^{\prime }=v_{y}+a_{y}\Delta t+\frac{1}{2}j_{y}\Delta t^{2}\\ a_{x}^{\prime }=a_{x}+j_{x}\Delta t\\ a_{y}^{\prime }=a_{y}+j_{y}\Delta t, \end{array}\right.$$

The motion model can be rewritten in its matrix representation as $x^{\prime }=Fx+\nu$, where
(7)$$F=\left(\begin{array}{cccccc} 1 & 0 & \Delta t & 0 & \frac{\Delta t^{2}}{2} & 0\\ 0 & 1 & 0 & \Delta t & 0 & \frac{\Delta t^{2}}{2}\\ 0 & 0 & 1 & 0 & \Delta t & 0\\ 0 & 0 & 0 & 1 & 0 & \Delta t\\ 0 & 0 & 0 & 0 & 1 & 0\\ 0 & 0 & 0 & 0 & 0 & 1 \end{array}\right)$$
(8)$$\left(\begin{array}{c} p_{x}^{\prime }\\ p_{y}^{\prime }\\ v_{x}^{\prime }\\ v_{y}^{\prime }\\ a_{x}^{\prime }\\ a_{y}^{\prime } \end{array}\right)=\left(\begin{array}{cccccc} 1 & 0 & \Delta t & 0 & \frac{\Delta t^{2}}{2} & 0\\ 0 & 1 & 0 & \Delta t & 0 & \frac{\Delta t^{2}}{2}\\ 0 & 0 & 1 & 0 & \Delta t & 0\\ 0 & 0 & 0 & 1 & 0 & \Delta t\\ 0 & 0 & 0 & 0 & 1 & 0\\ 0 & 0 & 0 & 0 & 0 & 1 \end{array}\right)\left(\begin{array}{c} p_{x}\\ p_{y}\\ v_{x}\\ v_{y}\\ a_{x}\\ a_{y} \end{array}\right)+\left(\begin{array}{c} \frac{1}{6}j_{x}\Delta t^{3}\\ \frac{1}{6}j_{y}\Delta t^{3}\\ \frac{1}{2}j_{x}\Delta t^{2}\\ \frac{1}{2}j_{y}\Delta t^{2}\\ j_{x}\Delta t\\ j_{y}\Delta t \end{array}\right)$$

The jerk could be expressed as the last term in the above equation. The random noise jerk vector ν is given by Equation (Equation 9) as: (9)$$\nu =\left(\begin{array}{c} \frac{1}{6}j_{x}\Delta t^{3}\\ \frac{1}{6}j_{y}\Delta t^{3}\\ \frac{1}{2}j_{x}\Delta t^{2}\\ \frac{1}{2}j_{y}\Delta t^{2}\\ j_{x}\Delta t\\ j_{y}\Delta t \end{array}\right)=\left(\begin{array}{cc} \frac{\Delta t^{3}}{6} & 0\\ 0 & \frac{\Delta t^{3}}{6}\\ \frac{\Delta t^{2}}{2} & 0\\ 0 & \frac{\Delta t^{2}}{2}\\ \Delta t & 0\\ 0 & \Delta t \end{array}\right)\left(\begin{array}{c} j_{x}\\ j_{y} \end{array}\right)=Gj$$

The process covariance matrix is Q, so ν∼N(0,Q). This covariance matrix is written by Equation (Equation 10) as: (10)$$Q=E\left[\nu \nu^{T}\right]=E\left[Gjj^{T}G^{T}\right]=GE\left[jj^{T}\right]G^{T}=G\left(\begin{array}{cc} \sigma_{jx}^{2} & 0\\ 0 & \sigma_{jy}^{2} \end{array}\right)G^{T}$$

After the matrix multiplication calculation, the process covariance matrix Q is given by Equation (Equation 11) as: (11)$$Q=\left(\begin{array}{cccccc} \frac{\Delta t^{6}\sigma_{jx}^{2}}{36} & 0 & \frac{\Delta t^{5}\sigma_{jx}^{2}}{12} & 0 & \frac{\Delta t^{4}\sigma_{jx}^{2}}{6} & 0\\ 0 & \frac{\Delta t^{6}\sigma_{jy}^{2}}{36} & 0 & \frac{\Delta t^{5}\sigma_{jy}^{2}}{12} & 0 & \frac{\Delta t^{4}\sigma_{jy}^{2}}{6}\\ \frac{\Delta t^{5}\sigma_{jx}^{2}}{12} & 0 & \frac{\Delta t^{4}\sigma_{jx}^{2}}{4} & 0 & \frac{\Delta t^{3}\sigma_{jx}^{2}}{2} & 0\\ 0 & \frac{\Delta t^{5}\sigma_{jy}^{2}}{12} & 0 & \frac{\Delta t^{4}\sigma_{jy}^{2}}{4} & 0 & \frac{\Delta t^{3}\sigma_{jy}^{2}}{2}\\ \frac{\Delta t^{4}\sigma_{jx}^{2}}{6} & 0 & \frac{\Delta t^{3}\sigma_{jx}^{2}}{2} & 0 & \Delta t^{2}\sigma_{jx}^{2} & 0\\ 0 & \frac{\Delta t^{4}\sigma_{jy}^{2}}{6} & 0 & \frac{\Delta t^{3}\sigma_{jy}^{2}}{2} & 0 & \Delta t^{2}\sigma_{jy}^{2} \end{array}\right)$$

The DWM1001’s LE was able to directly output the 3D coordinates, which enabled us to linearly map the state vector into the measurement space. Therefore, we could use the KF method for sensor fusion, which requires less computation compared with EKF or UKF. Here, *z* is defined as the measurement vector and H is the matrix that projects the current state vector into the measurement space.
(12)$$z=\left(\begin{array}{c} p_{x}\\ p_{y}\\ a_{x}\\ a_{y} \end{array}\right),H=\left(\begin{array}{cccccc} 1 & 0 & 0 & 0 & 0 & 0\\ 0 & 1 & 0 & 0 & 0 & 0\\ 0 & 0 & 0 & 0 & 1 & 0\\ 0 & 0 & 0 & 0 & 0 & 1 \end{array}\right)$$
(13)$$z=Hx^{\prime }+\omega ,$$
where ω is the measurement uncertainty with a measurement noise covariance matrix R as given by Equation (Equation 14). The parameters can be adjusted according to our confidence with the sensor’s measurements.   
(14)$$R=\left(\begin{array}{cccc} \sigma_{px}^{2} & 0 & 0 & 0\\ 0 & \sigma_{py}^{2} & 0 & 0\\ 0 & 0 & \sigma_{ax}^{2} & 0\\ 0 & 0 & 0 & \sigma_{ay}^{2} \end{array}\right),$$

KF contains loops of the prediction and update procedure. The matrix P is the state covariance matrix, which contains the uncertainty of the object’s position, velocity, and acceleration. All of the previously defined matrices are used in the following KF sensor fusion algorithm’s prediction and update procedures. The KF sensor fusion algorithm is shown in Algorithm 1 [45].
**Algorithm 1** Algorithm of the sensor fusion UWB and IMU.**Input:**px,py,ax,ay,timestamp**Output:** state vector *x* **while** true **do**     **if** is_initialized is false **then**         initialize vx,vy in the state vector *x* to zero         previous_timestamp←timestamp         is_initialized←true         continue     **end if**     Δt←timestamp−previous_timestamp     previous_timestamp←timestamp     calculate state transition matrix F using Δt     calculate process covariance matrix Q using Δt     x′←Fx                ▹ Prediction Procedure     P′←FPFT+Q     y←z−Hx′                ▹ Update Procedure     S←HP′HT+R     K←P′HTS−1     x←x′+Ky     P←(I−KH)P′ **end while**

### 3.4. Kalman Filter, EKF, and UKF Comparison

In this work, we used the DWM1001 which has a Nordic Semiconductor nRF52832 system-on-chip (SoC) onboard and supports PANS firmware. Nordic nRF52832 is based on a 64 MHz ARM Cortex M4 and it supports Bluetooth, UART, and SPI communication interfaces. Bluetooth enables DWM1001 to be configured wirelessly through an Android tablet. Furthermore, PANS firmware, which is built on an eCos real-time operating system (RTOS), can be configured to turn on the LE to directly output the location coordinates. This means that the location coordinates are calculated on DWM1001 instead of on the main computer. DWM1001 sends out these coordinates to the main computer at a refresh rate of 10 Hz. This shifts the coordinate calculation task from the main computer to DWM1001. It dramatically reduces the computation load on the main computer. Works from other research groups typically relied on distance measurements between tags and anchors. Thus, to calculate the coordinates, they would need to use either the trilateration method as Equation (Equation 2) or the classic least square (LS) method [46]. With the help of DWM1001, its PANS firmware supports LE and the coordinate calculation procedure is done on nRF52832. This is the major reason for selecting DWM1001 instead of DWM1000.

The EKF method was widely used in previous papers. They usually used the TWR method to measure distances between the tag and each anchor. However, the state vector, as shown in Equation (Equation 4), uses coordinates instead of distances. There is no H matrix as shown in Equation (Equation 12) that can linearly map the state vector into the distance measurements. If the mapping procedure is nonlinear, the Gaussian noise would not stay in the Gaussian distribution after the mapping. Thus, one would need to use the Taylor series expansion to find a linear approximation of the system. This results in calculating the Jacobian matrix $H_{j}$, which requires solving partial derivatives as given by Equation (Equation 15). EKF’s linear approximation procedure also introduces the linearization error.
(15)$$\left\{\begin{array}{c} \frac{\partial d_{i}}{\partial x}=\frac{x-x_{i}}{\sqrt{{\left(x-x_{i}\right)}^{2}+{\left(y-y_{i}\right)}^{2}}}\\ \frac{\partial d_{i}}{\partial y}=\frac{y-y_{i}}{\sqrt{{\left(x-x_{i}\right)}^{2}+{\left(y-y_{i}\right)}^{2}}}, \end{array}\right.$$

The UKF method faces the same nonlinear problem as the EKF method considering the measurements are distances. Instead of calculating the Jacobian matrix for linear approximation, UKF uses an unscented transformation (UT) to capture the nonlinear transformation. The posterior probability density distribution is obtained by carefully selecting a set of deterministic sample points and calculating the observation mean and the observation covariance matrix. These sample points are called sigma points. The selection of sigma points will influence the performance of the UKF algorithm. By generating sigma points, predicting sigma points, and calculating the observation mean and covariance matrix—all of these procedures require extra calculations on the main computer.

Compared with the previous EKF and UKF methods, the proposed KF method would require fewer computations as there is no need for the calculation of the Jacobian matrix, sigma point generation and prediction. Our system enables us to linearly map the state vector into the measurement space by using a simple H matrix as given by Equation (Equation 12). The proposed KF method reduces time complexity and suitable for deployment on a low-power embedded computer.

## 4. Experimental Results and Discussion

In real world situations, both the system motion and measurements are influenced by noise. If the noise follows a Gaussian distribution and the state vector can be linearly mapped into the measurement space, the KF sensor fusion method can be used to estimate the state of the system. Our testing platform and tracking system were placed in the same room. The UWB tag was surrounded by four UWB anchors, which were in LOS. Each dynamic test took less than 30 s. No dropout or delay was observed during our data logging procedure. The only assumption we made in our tracking system was that the uncertainty could be modeled as a Gaussian distribution.

The experiment setup required installing four anchors around our laboratory and their coordinates were configured manually by using an Android tablet running the Decawave DRTLS Manager. The anchors were placed at a height of 1.8 m and the distance between each anchor was measured by using a tape measure. The 1.8 m height ensured that the tag and four anchors were always in LOS to achieve the best performance. The Newmark CS-1000-1 linear stage and our tracking system were placed along the *y*-axis. For each test, it started from one end of the linear stage at coordinates (2.43, 1.00, 0.62) and moved to the other end at coordinates (2.43, 1.90, 0.62). The arrangements of the four anchors and the test platform are shown in Figure 5.

The target speed and acceleration time were directly configured in Newmark’s QuickMotion software. The CS-1000-1 linear stage accelerated at a constant acceleration, which could be calculated using the target speed divided by the acceleration time. Each dynamic test started from one end of the testing platform; the linear stage accelerated from static to its target speed with constant acceleration. After the target speed was reached, it remained in uniform motion with this target speed for a while. Then, it started to decelerate. The speed decelerated to zero when it reached the other end of the testing platform. We picked different combinations of acceleration and velocity as the ground truth to test our tracking system’s dynamic performance. The length of the CS-1000-1 linear stage was 1 m. A length of 5 cm at each end was reserved for safety operation. Therefore, there was only 90 cm of useful length for our testing. The tracking device was rigidly installed on the linear stage and moved back and forth from one end to the other end. This linear trajectory was considered our ground truth trajectory. The acceleration and velocity of the linear stage were considered our ground truth acceleration and velocity.

There were two different kinds of position errors on each axis: the root mean squared error (RMSE) and the maximum error. The maximum error on each axis was calculated by subtracting the ground truth value on that axis from the KF method-estimated location coordinates. The maximum values in those calculations are considered the maximum errors. The maximum error defines the worst localization performance on the axis. RMSE defines the average localization performance and is calculated by Equation (Equation 15) [47].
(16)$$RMSE=\sqrt{\frac{1}{n}\sum_{t=1}^{n}{\left(x_{t}^{est}-x_{t}^{true}\right)}^{2}},$$

### 4.1. UWB LE Performance on the *z*-axis without the KF Sensor Fusion

The purpose of this section is to show that UWB LE localization performance degraded when increasing the target speed and accelerating the testing platform. There was no sensor fusion algorithm applied on the *z*-axis. Location data on the *z*-axis came directly from UWB LE; these were raw data, which only indicated UWB’s localization performance under dynamic test conditions.

Considering that the testing platform can only move in a 2D xy-plane and the KF was designed on a 2D xy-plane, the data on the *z*-axis should be analyzed separately. Because there is no movement on the *z*-axis during the dynamic testing procedure, theoretically, the UWB LE measurements on the *z*-axis should remain constant. However, both the noise and the movements on the *x*-axis and *y*-axis could influence the location measurements on the *z*-axis. According to our dynamic test results, UWB LE data on the *z*-axis were influenced when the testing platform was in motion. The experiment results show a maximum error of up to 50 cm on the *z*-axis during the linear stage’s acceleration and deceleration along the *y*-axis. UWB LE raw data measurements on the *z*-axis are shown in Figure 6. This can be explained by Equation (Equation 2). Because UWB LE needs to use the trilateration method to calculate the coordinates from the measured distances, the x, y, and z coordinates are inherently coupled to each other.

The consumer-grade IMU BNO055 also measured noisy data on the *z*-axis when the linear stage accelerated and decelerated along the *y*-axis. This is shown in Figure 7. Although its performance may be acceptable for applications that do not require highly accurate or reliable location services, it is better to use a higher-grade IMU.

The *z*-axis is reserved for evaluating localization performance by only using a UWB sensor under dynamic testing conditions. The maximum error and RMSE on the *z*-axis under different dynamic testing conditions are listed in Table 4. For each combination of target speed and acceleration, the same testing was repeated five times and the average result was calculated to eliminate random error. Let us first analyze the relationship between acceleration and location error. The target speed was set to 600 mm/s, as shown in row 1 and row 4, when the acceleration increased from 0.6 m/s^2^ to 1.2 m/s^2^; both RMSE and maximum error increased. The target speed was set to 800 mm/s, as shown in row 3 and row 6; both RMSE and maximum error also increased when the acceleration increased from 0.8 m/s^2^ to 1.6 m/s^2^. The target speed was set to 700 mm/s as shown in row 2 and row 5; when the acceleration increased from 0.7 m/s^2^ to 1.4 m/s^2^, and the maximum error still increased, but the RMSE decreased. Although there is no established rule that states that the location error would increase with acceleration, we observe that the UWB’s localization performance tends to degrade when the acceleration increases.

The relationship among target speed, RMSE, and maximum error is also evaluated on the linear stage testing platform. The acceleration was set to a constant value when increasing the linear stage’s target speed. Each test started from zero speed, accelerated to the target speed, then decelerated to zero. The displacement traveling from zero to the target speed can be calculated by using the equation $s=v^{2}/2a$. The minimum length required to perform the experiment should double this equation’s calculation considering the deceleration also needed to travel the same displacement. If we want to achieve the same target speed, a smaller acceleration setting would result in traveling a longer distance. However, the useful length of our testing platform is 90 cm, so we chose a relatively high acceleration of 1.6 m/s^2^ and held this value constant for the following test. The target speeds varied between 600, 700, and 800 mm/s; the RMSE and the maximum error are listed in Table 5. Although this may not be an established rule, we observe that the UWB’s localization performance tends to degrade with the increasing target speed.

The ground truth coordinate on the *z*-axis was 0.62 m. The testing platform only accelerated and decelerated along the *y*-axis and there was no motion along the *z*-axis. The localization performance of UWB under stationary conditions worked well. However, it degraded quickly when the testing platform was in motion, as shown in Figure 8, which is a hypothesis of this paper. If there is motion, no matter what direction of that motion, it can cause the UWB localization performance to degrade. A higher target speed and a higher acceleration will result in a worse UWB localization performance. When the testing platform is stationary, the location variance of UWB is small, as mentioned before. It is when there is movement under dynamic testing conditions that the variance increases.

### 4.2. Localization Performance on the X-Axis and Y-Axis with the KF Sensor Fusion

When using the KF sensor fusion method, the variance of a sensor is usually provided on its datasheet and this value can be used to determine the parameters in the measurement noise covariance matrix R as given by Equation (Equation 14). By adjusting the entries in matrix R, we can adjust our confidence level to match a certain sensor’s measurements. In this matrix, the first and second rows were UWB variances, the third and fourth rows were BNO055 variances. A higher variance means less confidence in that sensor’s measurements. The UWB variance was set to a small number, considering the tag and four anchors were in good LOS. The measurements of BNO055 were quite noisy, which required setting its variance to a relatively higher value. Different configurations of IMU variances result in different localization performances. Here, we picked 10 IMU variance settings from 0.1 to 1 for BNO055. Then, these configurations were applied to the same KF sensor fusion algorithm (shown in Algorithm 1) to calculate the location estimation. Finally, both maximum error and RMSE were calculated. These errors were plotted against IMU variance, which is shown in Figure 9. Both maximum error and RMSE decreased with increasing IMU variance.

To provide a more intuitive idea about how different IMU variance settings can influence the localization performance, we picked two different measurement noise covariance matrices, $R_{1}$ and $R_{2}$, as given by Equation (Equation 17). Both $R_{1}$ and $R_{2}$ have the same entries for the UWB variance, but entries in $R_{2}$ for the IMU variance have smaller numbers, which means $R_{2}$ places more weight on the IMU measurements. The KF trajectory comparison is shown in Figure 10. Considering the linear stage only moves along the *y*-axis, there should be no motion along the *x*-axis. The result shows that the KF trajectory becomes worse when placing more weight on IMU measurements. These experimental results show that a higher-grade IMU is needed when a more reliable and accurate location service is required.
(17)$$R_{1}=\left(\begin{array}{cccc} 0.0225 & 0 & 0 & 0\\ 0 & 0.0225 & 0 & 0\\ 0 & 0 & 1 & 0\\ 0 & 0 & 0 & 1 \end{array}\right),R_{2}=\left(\begin{array}{cccc} 0.0225 & 0 & 0 & 0\\ 0 & 0.0225 & 0 & 0\\ 0 & 0 & 0.2 & 0\\ 0 & 0 & 0 & 0.2 \end{array}\right),$$

A low-cost consumer-grade IMU BNO055 was selected for this project as a proof-of-concept. The noise and biases from BNO055 (when integrated over time) can cause large position errors. BNO055 acceleration measurements under the motion only accelerate and decelerate along the *y*-axis, as shown in Figure 11. For better localization results, BNO055 can be replaced with a more expensive and higher-performance IMU.

The values of x, y coordinates are shown in Figure 12. The KF sensor fusion algorithm is applied to the xy-plane. Considering the linear stage only accelerates and decelerates along the *y*-axis, there should be no movements along the *x*-axis on the ground truth trajectory. The linear stage testing platform was configured to move back and forth from one end to the other end along the *y*-axis; the distance it traveled was 90 cm. The ground truth on the *x*-axis was 2.43 m (and should remain constant during dynamic testing). The RMSE and maximum error could be calculated conveniently on the *x*-axis. The RMSE was only 6 cm and the maximum error was 13.7 cm on the *x*-axis. Considering the y coordinates kept changing under the dynamic testing conditions, we only compared the traveled distance on the *y*-axis. Theoretically, the ground truth traveled distance should be 0.9 m from one end to the other end. However, the experimental result was around 0.83 m. Thus, the error on the *y*-axis was considered to be 7 cm. Our dynamic testing results are very close to those from previous works, which are shown in Table 1. In conclusion, the KF sensor fusion of UWB and IMU can achieve an accuracy of less than 15 cm under dynamic testing conditions.

The length of the testing platform was limited to 1 m considering the cost of a linear stage increases dramatically with its length. The major reason to use a linear stage is that both its target speed and acceleration can be controlled very accurately. The linear stage provides highly accurate ground truth trajectory, which is required to evaluate centimeter-level localization performances. However, the 1 m length limitation may raise some concerns about whether the testing result is useful in real-world indoor positioning applications. We believe it is useful as the 1 m limitation only applies to the tracking device and not to the anchor’s installation. Secondly, the maximum location error of the proposed KF method is 13.7 cm. A 1 m length may not be ideal, but it is sufficient considering the location error is low. Lastly, the complete trajectory (when moving in a large area) could be divided into many small unit trajectories; each unit trajectory could comply with our 1 m testing results as long as the tag is in UWB’s operating range, and the tag is in LOS. In a more realistic indoor positioning scenario, the tag moves around a large area and the localization performance is expected to become worse when the tag moves far away from the anchors. However, as long as the tag is still in UWB’s operating range, this paper’s experimental results can still serve as a good reference for designing a positioning system.

Our testing platform also contained a motorized rotary stage RM-3-110. The rotations around the x, y, z axes were captured by IMU’s gyroscope. The orientation of the testing platform will be investigated in future research.

## 5. Conclusions and Future Work

We built a low-cost, portable, tracking system hardware using a Raspberry Pi 4 computer, DWM1001 UWB, and BNO055 IMU. The data logging software was developed using C programming language and runs on a Linux operating system. The KF sensor fusion algorithm was developed in Matlab to process the logged data. UWB LE localization performance is good when the testing platform is static. However, by increasing the target speed and acceleration, UWB LE location data start to become worse. By adding an IMU and applying the KF sensor fusion algorithm, it achieves better localization results compared with only relying on a UWB sensor.

The dynamic performance of the tracking system was tested on a Newmark CS-1000-1 linear stage, which can generate highly accurate ground truth trajectory, acceleration, and velocity. The experimental results show that the localization accuracy of UWB will become worse by increasing the target velocity and acceleration of the testing platform. This performance degradation can be mitigated by applying the KF sensor fusion method with an IMU.

This work focuses on the dynamic performance of the UWB/IMU tracking system. Here, we ignored the influence of many external factors on location errors by setting the experiment in an ideal laboratory environment. First, non-line-of-sight (NLOS) testing conditions will increase the location error. Our tracking system is installed in a good LOS to avoid the influence of NLOS. Second, the distance between the anchor and tag will influence the localization performance. If the distance is too long, the location error will increase. The height of the anchor’s installation will influence the location error. Finally, multipath, signal reflection, and RF interference could also contribute to the location error.

In future work, a higher grade IMU will be used to develop our next tracking system. The testing platform will be upgraded to a robotic arm, which could generate a 3D trajectory, acceleration, and velocity. The Kalman filter would also be redesigned in 3D. This research could be applied to autonomous systems which navigate in GPS-denied environments and require high localization accuracy.

## Figures and Tables

**Figure 1 sensors-22-08156-f001:**
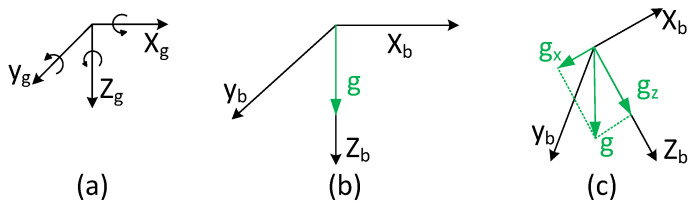
Gravitational acceleration and different coordinate frames: (**a**) Global frame. (**b**) Body frame without rotation. (**c**) Body frame rotation around the *y*-axis.

**Figure 2 sensors-22-08156-f002:**
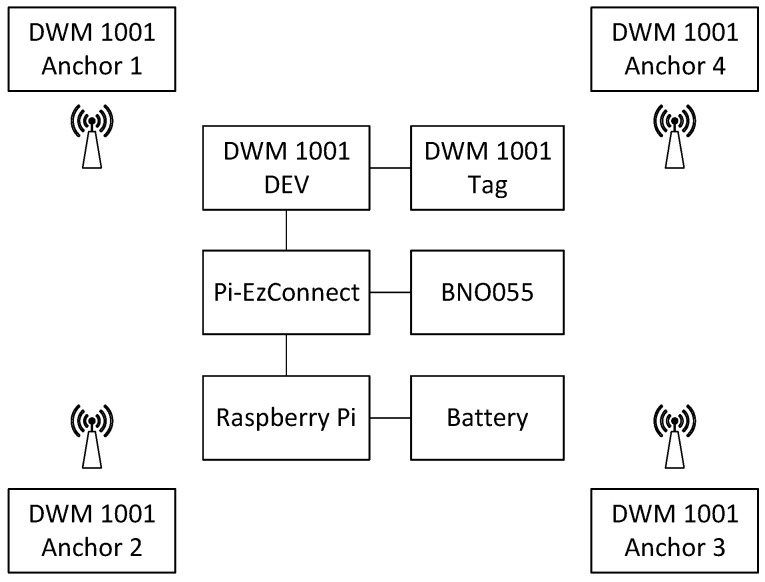
System design of the UWB and IMU tracking system.

**Figure 3 sensors-22-08156-f003:**
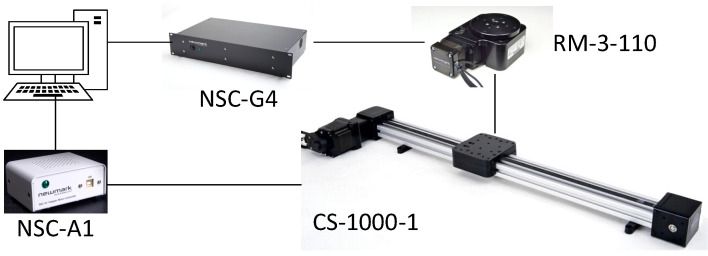
The testing platform setup.

**Figure 4 sensors-22-08156-f004:**
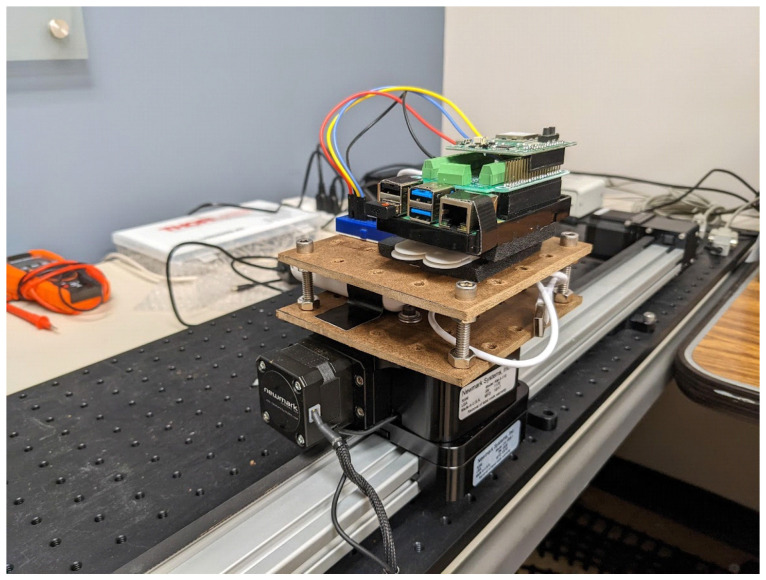
The UWB/IMU tracking device installed on the testing platform.

**Figure 5 sensors-22-08156-f005:**
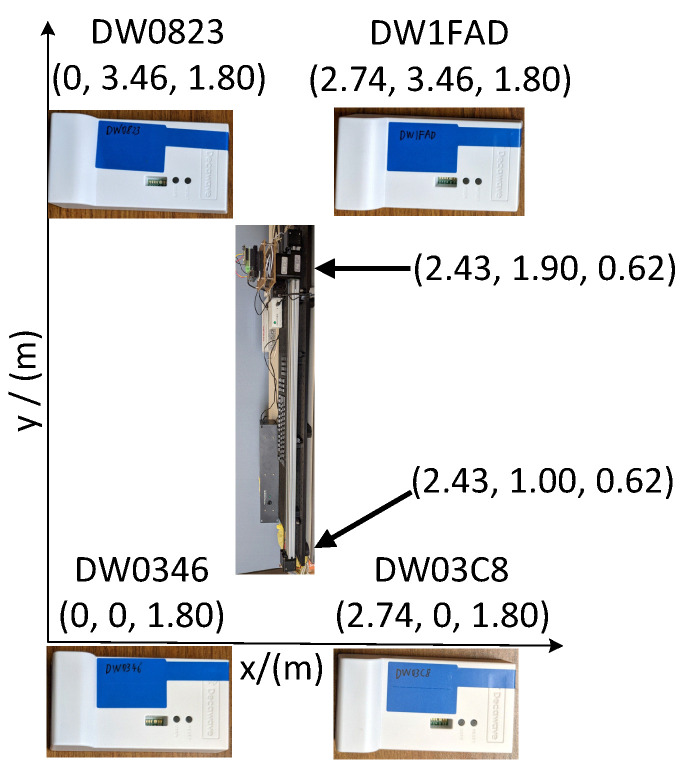
The arrangement of the four anchors and the test platform.

**Figure 6 sensors-22-08156-f006:**
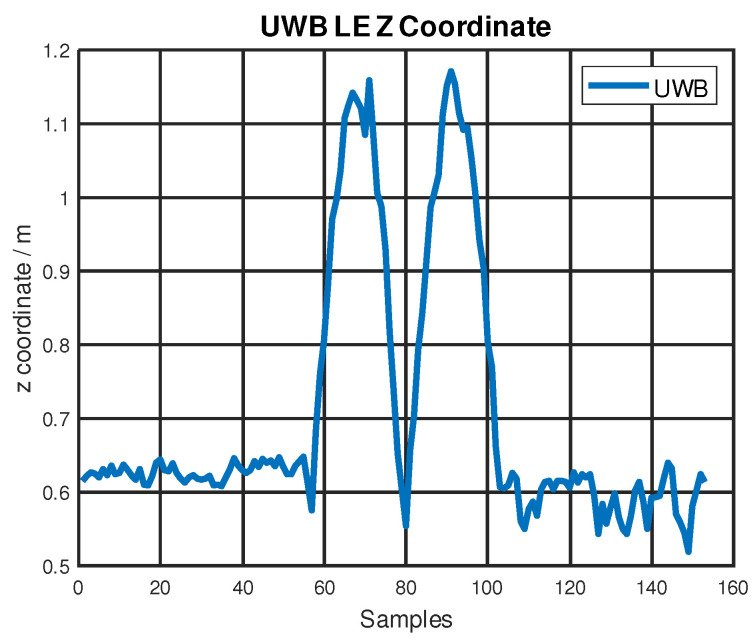
UWB LE raw data measurements on the *z*-axis during dynamic testing.

**Figure 7 sensors-22-08156-f007:**
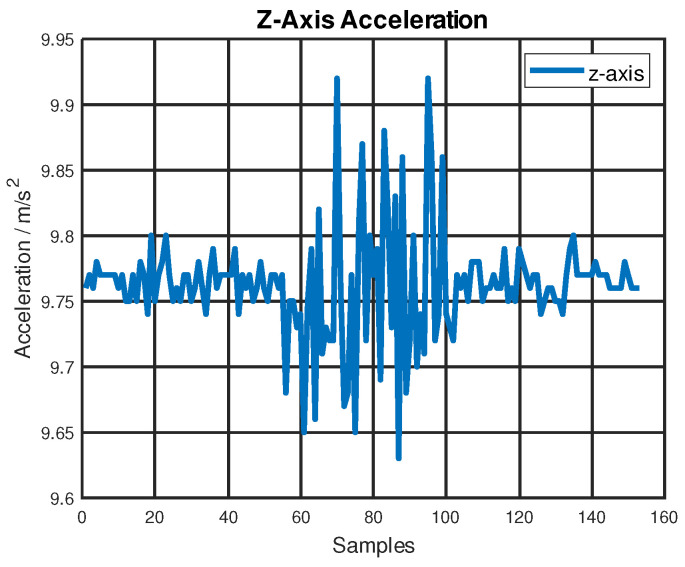
BNO055 acceleration measurements on the *z*-axis during dynamic testing.

**Figure 8 sensors-22-08156-f008:**
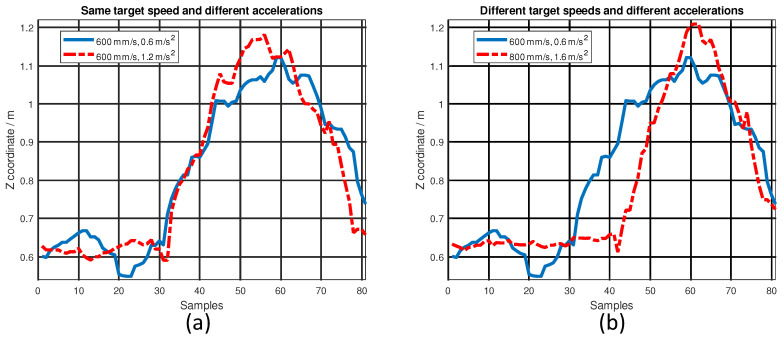
*z*-axis coordinates under: (**a**) the same target speed and different accelerations; (**b**) different target speeds and different accelerations.

**Figure 9 sensors-22-08156-f009:**
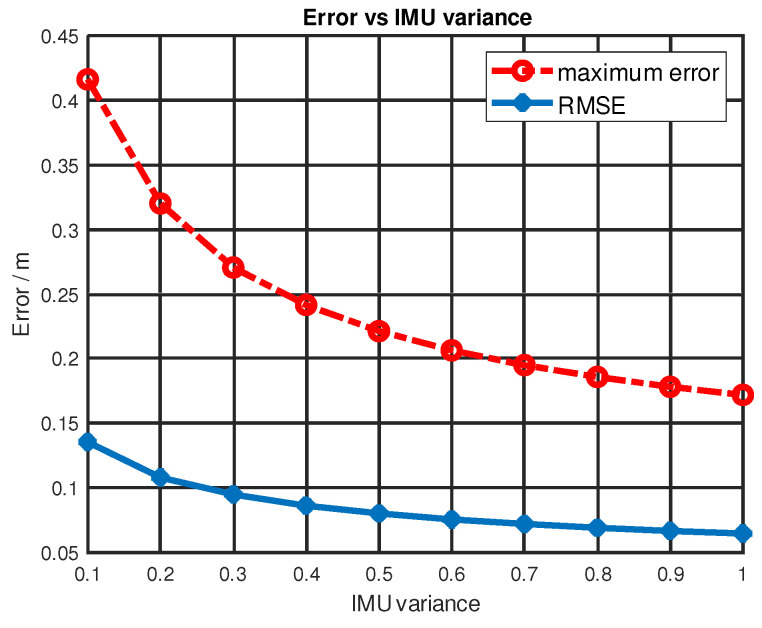
Maximum error and RMSE vs. IMU variance.

**Figure 10 sensors-22-08156-f010:**
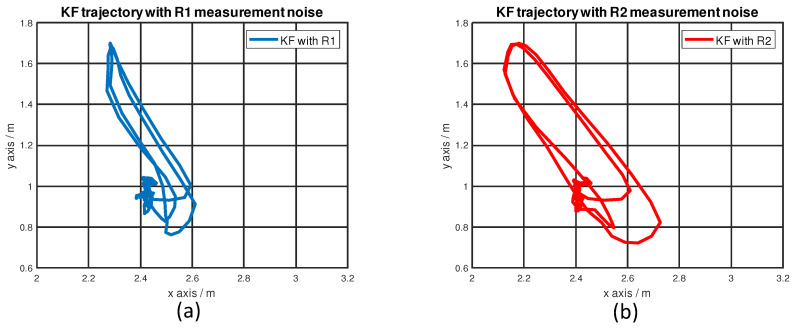
KF trajectory comparison: (**a**) $R_{1}$ measurement noise covariance matrix; (**b**) $R_{2}$ measurement noise covariance matrix.

**Figure 11 sensors-22-08156-f011:**
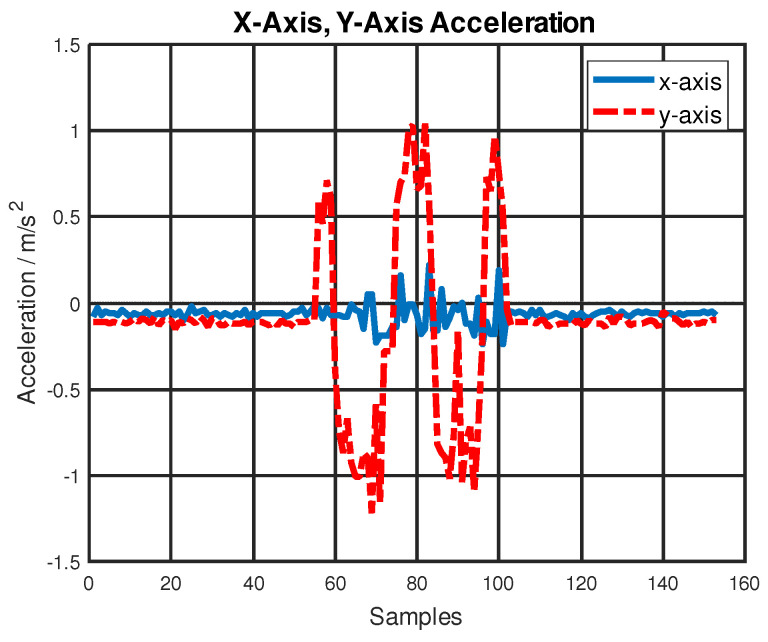
BNO055 acceleration data on the *x*-axis and *y*-axis.

**Figure 12 sensors-22-08156-f012:**
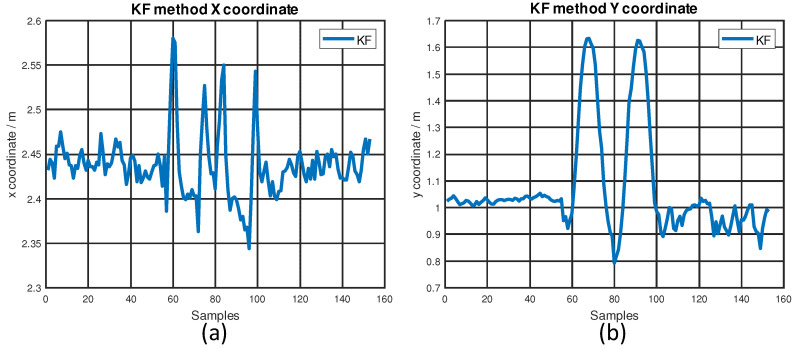
The values of x, y coordinates: (**a**) x coordinate; (**b**) y coordinate.

**Table 1 sensors-22-08156-t001:** Comparison of prior works using UWB and IMU localization.

Method	Accuracy	References	Year
KF	0.14 m	[14]	2008
EKF	0.03 m	[15]	2015
EKF	0.20 m	[16]	2015
EKF	0.18 m	[17]	2017
KF	0.36 m	[18]	2017
EKF	0.39 m	[19]	2018
EKF/UKF	0.08 m	[20]	2020
KF	0.16 m	[21]	2021
Adaptive EKF	0.19 m	[22]	2022
EKF	0.21 m	[23]	2022

**Table 2 sensors-22-08156-t002:** Materials used to build the tracking system and test platform.

Materials	Description
UWB	Decawave MDEK1001
IMU	BNO055
Computer for Tracking System	Raspberry Pi 4 Model B 8GB RAM
Computer for Test Platform	HP ProDesk Intel Core i5
Tablet	SAMSUNG Galaxy Tab A7
Raspberry Pi Shield Board	Pi-EzConnect
Power Bank	Adafruit USB Li-Ion Power Bank
Battery	RCR123A
Linear Stage	Newmark CS-1000-1
Rotary Stage	Newmark RM-3-110
Linear Motion Controller	Newmark NSC-A1
Rotary Motion Controller	Newmark NSC-G4

**Table 3 sensors-22-08156-t003:** Newmark CS-1000-1 belt drive’s linear stage specifications.

Parameter	Description
Travel Range	1000 mm
Resolution	3.6 μm (@125 microsteps)
Encoder	Optical rotary encoder, 4000 CPR with index
Linear Travel Per Motor Revolution	90.0104 mm/rev
Uni-directional Repeatability	20 μm
Bi-directional Repeatability	80 μm
Maximum Speed	1 m/s
Maximum Load	4.5 kg

**Table 4 sensors-22-08156-t004:** The maximum error and RMSE on the *z*-axis under different dynamic testing conditions.

Target Speed, mm/s	Acceleration, m/s^2^	RMSE, m	Maximum Error, m
600	0.6	0.334	0.494
700	0.7	0.372	0.567
800	0.8	0.349	0.544
600	1.2	0.382	0.554
700	1.4	0.360	0.573
800	1.6	0.374	0.593

**Table 5 sensors-22-08156-t005:** The maximum error and RMSE on the *z*-axis under the same acceleration and different target speeds.

Target Speed, mm/s	Acceleration, m/s^2^	RMSE, m	Maximum Error, m
600	1.6	0.364	0.569
700	1.6	0.368	0.582
800	1.6	0.374	0.593

## Data Availability

The data presented in this research is available upon request from C.L.

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
