# Peer review of "Experimental Evaluation of Sensor Fusion of Low-Cost UWB and IMU for Localization under Indoor Dynamic Testing Conditions"

_sensors, 2022, doi:10.3390/s22218156_

Round 1

Reviewer 1 Report

The Authors discuss the problem of the influence of object movement on positioning accuracy in UWB systems. The key contribution is the experimental assessment of the positioning accuracy in a controlled environment. A dedicated measurement setup has been dsigned for this purpose - forcing the object to move at some desired speed/acceleration.

The positioning itself is done using UWB Decawave development boards/modules. Then, the time of flight analysis is performed to estimate the distance between the moving object and the reference UWB transceivers (positioning anchors). The Authors show the reference results obtained in the test environment for different motion parameters. Then, the Authors propose the use of off-the-shelf IMU modules to collect additional data for improvement of positioning accuracy. The IMU supported setup allows to improve object tracking accuracy in indoor scenario.

The paper is well written and organized. The contribution of the Authors is celarly stated (experimental evaluation of the positioning accuracy in UWB systems, design of an original measurement testbed, implementation and evaluation of IMU supported positioning). The results are sufficeitnly well documented, and the conclusion supported with the results.

However, the paper leaves some space for improvement. Mainly:

- the travel range of the belt drive of the linear stage in the measurement setup is limited to 1 m only - this seems to be a relatively short distance compared to the dimensions of a typical indoor environment. There is no discussion on how this limit affects the applicability of the results to more realistic indoor positioning scenarions.

- what was the expected influence of measurement environment on the positionign accuracy (i.e. configuration of walls reflecting the EM waves, and the layout of the anchors)?

- Figure 5 - the coordinates of the test platform have not been provided.

Minor issues:

Table 2

Rotary stage -> Rotary Stage (capitalization as in other rows)

Reviewer 2 Report

The novel contribution in this work is the low-cost, low-power tracking systems hardware, software design and the experiment setup to observe the tracking systems localization performance under different dynamic testing conditions. I think there are some major issues in the manuscript that need to be addressed, some of them are listed below.

[1] IMU and UWB have been widely used in indoor positioning systems, so it is recommended that the authors add some state-of-the-art indoor positioning algorithms based on IMU and UWB as comparison baselines, which has demonstrated the advantages of this manuscript.

[2]One of the main innovations of this paper is that KF method is used for sensor fusion of UWB and IMU, which can reduce the time complexity compared with EKF or UKF method. It is suggested that the authors analyze the time complexity of program operation through experiments, and whether the positioning error is increased or decreased compared with EKF and UKF.

[3]The main innovation of this manuscript is the design of experimental system, which establishes a low-cost, portable indoor position tracking system. However, this is not enough. Compared with the existing similar methods, the main theoretical contribution of this paper is not very clear.

[4]In lines 259 to 265, the authors describe that in a real world, both motion and measurement are influenced by noise. If the noise follows a Gaussian distribution and the state vector can be linearly mapped into the measurement space, KF sensor fusion method can be used to estimate the state of the system. The only assumption made by the authors in the tracking system is that the uncertainty can be modeled as Gaussian distribution. Does it mean that the method proposed in this paper can only be applied to Gaussian noise models.

[5] In lines 317 to 318, the authors describe that UWBs localization performance has a tendency to degrade with increasing the velocity and acceleration. However, it can be seen from the numerical results in Table 4 that the root mean square error and maximum error of UWB positioning do not increase with the increase of velocity and acceleration.

Round 2

Reviewer 2 Report

The authors have answered my comments, acceptance is suggested.